# Numerical Analysis on Erosion and Optimization of a Blast Furnace Main Trough

**DOI:** 10.3390/ma14174851

**Published:** 2021-08-26

**Authors:** Hao Yao, Huiting Chen, Yao Ge, Han Wei, Ying Li, Henrik Saxén, Xuebin Wang, Yaowei Yu

**Affiliations:** 1State Key Laboratory of Advanced Special Steel, Shanghai Key Laboratory of Advanced Ferrometallurgy, Department of Materials Engineering, School of Materials Science and Engineering, Shanghai University, Shanghai 200444, China; yaohao159123@shu.edu.cn (H.Y.); huitingchen@shu.edu.cn (H.C.); ge_geyao@163.com (Y.G.); weihan@shu.edu.cn (H.W.); yingli@shu.edu.cn (Y.L.); 2Process and Systems Engineering Laboratory, Faculty of Science and Engineering, Åbo Akademi University, Henriksgatan 2, 20500 Abo, Finland; henrik.saxen@abo.fi; 3LaiSteel Technology Center, Laiwu Iron & Steel Co., LTD, Laiwu 271104, China; erli2000@126.com

**Keywords:** main trough, thermal stress, fatigue life, refractory, OpenFOAM

## Abstract

The main trough of a blast furnace (BF) is a main passage for hot metal and molten slag transportation from the taphole to the torpedo and the slag handling. Its appropriate working status and controlled erosion ensure a safe, stable, high-efficiency and low-cost continuous production of hot metal. In this work, the tapping process of a main trough of a BF in the east of China was numerically studied with the help of a CFD library written in C++, called OpenFOAM, based on the use of the Finite Volume Method (FVM). The results show that turbulence intensity downstream of the hot metal impact position becomes weaker and the turbulence area becomes larger in the main trough. During the tapping, thermal stress of wall refractory reaches the maximum value of $1{.7\;\times \;10}^{7}$ Pa at the 4 m position in the main trough. Furthermore, baffles in the main trough placed between 5.8 m and 6.2 m were found to control and reduce the impact of the turbulence on the refractory life. The metal flowrate upstream of the baffles can be decreased by 6%, and the flow velocity on the upper sidewall and bottom wall decrease by 9% and 7%, respectively, compared with the base model. By using baffles, the minimum fatigue life of the refractory in the main trough increases by 15 tappings compared with the base model, so the period between the maintenance stops can be prolonged by about 2 days.

## 1. Introduction

The main trough of a blast furnace (BF) is the only runner for hot metal and slag leaving the taphole and reaching the torpedo and slag handling unit, and an important place to separate slag from hot metal and desulfurize hot metal as well. The production rate and efficiency of the BF, hot metal quality and cost are affected directly by its working condition. With the large scale and long life length of the BF, the working intensity of the taphole is increasing [1]. About 4–7.5 t/min of hot metal and molten slag at 1730–1800 K [2] pass through the main trough. Tapping lasts for 70–120 min and there are typically 10–14 taps per day. In high-intensity tapping operation, the main trough consumes most refractory materials in the BF system [3,4]. At the end of a BF’s campaign, the accumulated cost of lining and repairing of the trough system exceeds the sum of new lining and all auxiliary equipment costs of the BF. Therefore, studying the working status of the main trough and clarifying the erosion mechanisms of the refractory can help prolong the life of the main trough lining and save costs in hot metal production.

At present, the primary research methods of the main trough can be classified as hydraulic model experiments, industrial tests and numerical calculations, while the main research has been focused on tapping parameters, properties and structure of the main trough and their effect on the liquid flow in the main trough. By analyzing hot metal in the main trough with the Fluent software, Chang et al. [5] found that the wall shear stress increases with the increase in taphole angle. The metal outflow rate has an impact on the working state of the main trough. Luo et al. [6] argued that the turbulence level in the trough gradually increases with the flow rate at the taphole and hot metal in slag is clearly observed. Furthermore, the shape of the trough has an obvious influence on the internal flow field, temperature field, refractory temperature field and slag–iron separation. Kim et al. [7] analyzed the influence of the length and cross-sectional area of the main trough on the separation of slag and hot metal in it by hydraulic model experiments and found that the effectiveness of slag–iron separation decreases if the main trough is short. Additionally, the influence of the trough length on slag–iron separation is less than the effect of the cross-sectional area. Dash et al. [8] used Fluent software to analyze the flow field near the skimmer in the main trough, and turbulence was found to appear near this position. The shear stress on the wall increases with the angle of the skimmer dam. Additionally, using Fluent, Kou et al. [9] concluded that the performance of slag–iron separation can be optimized by reducing the slope of the trough and increasing the width of it. Sun et al. [10] carried out hydraulic model experiments to analyze the flow state of the main trough with different slopes and found that a smaller slope of the trough gives a longer replacement time of hot metal, fewer temperature swings in the trough and less damage caused by temperature changes. Moreover, the relative position of the slag–iron trough (the position in front of skimmer) directly affects the service life of the skimmer. Li et al. [11] studied the influence of different slag port heights and skimmer height on slag–iron separation. The results indicated that the content of hot metal in the slag can be reduced by increasing the height of the lower opening of the skimmer and by reducing the tapping rate. Antonov et al. [12] adjusted the distance between the slag outlet and the skimmer and the width of the slag outlet to influence the internal flow field of the trough. On the basis of Fluent simulation and hydraulic model experiments, Luomala et al. [13] studied and optimized the flow field of the main trough by inserting a box into it.

In summary, based on the above research, turbulence of hot metal in the main trough is related to the falling hot metal. Furthermore, even if a flow controller is installed at the bottom of the main trough, the turbulence cannot be completely suppressed, but the flow velocities decrease. However, the working state of the main trough at a high temperature, as well as the coupling of hot metal and refractory heat transfer are not considered in the earlier simulation studies of the trough. Moreover, neither thermal stress distribution in the refractory or fatigue life of the refractory has been reported.

The present work computationally studies the tapping process of the main trough of No. 2 BF in Lai Steel, (Laiwu, China) with the help of the OpenFOAM code. Based on a three-dimensional model of the hot metal and refractory in the working layer, a numerical model of the main trough under the coupling of multiple physical fields is built. After studying the flow, temperature, pressure and thermal stress in the trough during tapping, the fatigue life of the refractory materials is predicted. Finally, by changing the structure of the main trough, reasonable estimates for optimal fatigue life of the main trough are suggested.

## 2. Mathematical Approach

The mathematical model is divided into three parts: hot metal and thermal properties, heat transfer and thermal stress in the working layer of the refractory material, and conjugated heat transfer between hot metal and the refractory.

### 2.1. Governing Equations of Hot Metal

#### 2.1.1. Mass Conservation

The governing equations solved in the hot metal include the continuity equation, momentum equation and energy equation. Assuming hot metal to be an incompressible Newtonian fluid, its volume expansion rate equals zero [14] so
(1)$$\frac{\partial u_{i}}{\partial x_{i}}=0,$$
where $x_{i}$ is the three-dimensional coordinate of a point (m) (where $x_{1}$, $x_{2}$ and $x_{3}$ represent the three orthogonal directions) and $u_{i}$ is the velocity corresponding to the coordinate point ($m\cdot s^{-1}$).

#### 2.1.2. Momentum Conservation

Substituting the constitutive equation of a Newtonian fluid into the dynamic equation, the momentum conservation equation of the incompressible Newtonian fluid is obtained [14] as
(2)$$\frac{\partial u_{i}}{\partial t}{+u}_{j}\frac{\partial u_{i}}{\partial x_{j}}=-\frac{1}{\rho }\frac{\partial p}{\partial x_{i}}+\nu \frac{\partial }{\partial x_{j}}(\frac{\partial u_{i}}{\partial x_{j}}),$$
where p is the pressure ($\mathrm{kg}\cdot m^{-1}\cdot s^{-2}$), *t* is the time (s), *ρ* expresses the density ($\mathrm{kg}\cdot m^{-3}$), and ν is the dynamic viscosity of the fluid ($m^{2}\cdot s^{-1}$).

The standard *k*-ε turbulence model has few empirical parameters, a simple form of calculation, usually gives accurate results and has a wide application range. Therefore, turbulence is calculated by the standard *k*-ε turbulence equations
(3)$$\nu_{t\;}{=C}_{\mu }\frac{k^{2}}{\varepsilon },$$
(4)$$\frac{\partial \left(\rho k\right)}{\partial t}+\frac{\partial \left({\rho ku}_{i}\right)}{\partial x_{i}}=\frac{\partial }{\partial x_{j}}\left[\left(\mu +\frac{\nu_{t}}{\sigma_{k}}\right)\frac{\partial k}{\partial x_{j}}\right]{+G}_{k}-\;\rho \varepsilon ,$$
(5)$$\frac{\partial \left(\rho \varepsilon \right)}{\partial t}+\frac{\partial \left({\rho \varepsilon u}_{i}\right)}{\partial x_{j}}=\frac{\partial }{\partial x_{j}}\left[\left(\mu +\frac{\nu_{t}}{\sigma_{\varepsilon }}\right)\frac{\partial \varepsilon }{\partial x_{j}}\right]+\frac{C_{1\varepsilon }\varepsilon }{k}G_{k}-{\;C}_{2\varepsilon }\rho \frac{\varepsilon^{2}}{k},$$
(6)$$G_{k}{=\nu }_{t}\left(\frac{\partial u_{i}}{\partial u_{j}}+\frac{\partial u_{j}}{\partial u_{i}}\right)\frac{\partial u_{i}}{\partial x_{j}},$$
where $C_{\mu }=0.09$, $C_{1\varepsilon }=1.44$, $C_{2\varepsilon }=1.92$, $\sigma_{k}=1.00$ and $\sigma_{\varepsilon }=1.30$.

#### 2.1.3. Energy Conservation

The energy conservation is expressed by [14]:(7)$$\frac{\partial T_{f}}{\partial t}+\frac{\partial u_{i}T_{f}}{\partial x_{i}}=\frac{\partial }{\partial x_{i}}\frac{K_{f}}{{\rho C}_{pf}}\frac{\partial T_{f}}{\partial x_{i}},$$
where $K_{f}$ is the thermal conductivity of hot metal ($W\cdot m^{-1}\cdot K^{-1}$), and $C_{pf}$ is the specific heat capacity of hot metal ($J\cdot {\mathrm{kg}}^{-1}\cdot K^{-1}$).

#### 2.1.4. Wall Shear Stress

For a further study on the mechanical erosion, wall shear stress is introduced to describe the erosion caused by hot metal on the wall. According to Newton’s law of viscosity [11,15,16], the wall shear stress of turbulence is
(8)$$\tau_{w}=-\left({\mu +\mu }_{t}\right)\;\times \;\rho \;\times \;\frac{\vec{u}}{\;\vec{n}},$$
where $\frac{\vec{u}}{\vec{n}}$ expresses the velocity gradient perpendicular to the wall, μ and ρ define the viscosity and density of hot metal, respectively, and $\mu_{t}$ is the turbulent viscosity.

### 2.2. Governing Equations of Refractory

#### 2.2.1. Fourier Equation

The heat conduction in the working layer of the refractories is described by the Fourier equation [17]
(9)$$\frac{\partial T_{f}}{\partial t}=\frac{\partial }{\partial x_{i}}\frac{K_{s}}{{\rho C}_{ps}}\frac{\partial T_{f}}{\partial x_{i}},$$
where ${\;K}_{s}$ and $C_{ps}$ define the thermal conductivity and specific heat capacity of the refractory material of the working layer, respectively ($W\cdot m^{-1}\cdot K^{-1}$) and ($J\cdot {\mathrm{kg}}^{-1}\cdot K^{-1}$).

The equations are solved by OpenFOAM with a displacement method. However, the displacement nodes are considered unknown, by which stress, strain and node force are expressed. The theory of thermal elasticity is similar to the elasticity theory, which establishes the basic equation of the elastomer problem with three aspects of dynamics, physics and geometry [18]. The following are thermal stress calculation equations of the main trough applied in this study.

#### 2.2.2. Momentum balance

The momentum balance is given by
(10)$$\frac{\partial^{2}\left(\rho_{s}D\right)}{\partial t^{2}}-\frac{\partial \sigma_{i}}{\;\partial x_{i}}=0,$$
where *D* is the displacement vector (m), $\rho_{s}$ is the density (kg·m^−3^) and *σ* is the stress tensor (Pa) of the refractory material, respectively. *σ* has nine components, where $\sigma_{x}{,\sigma }_{y}{,\sigma }_{z}$ express the normal stress (Pa) in the three directions in rectangular coordinates, $\tau_{xy}{,\tau }_{yx}{,\tau }_{yz}{,\tau }_{zy}{,\tau }_{xz}{\;\mathrm{and}\;\tau }_{zx}$ define the shear stress in six directions (Pa) and $\frac{\partial^{2}\left(\rho D\right)}{\partial t^{2}}$ denotes the volume force.

#### 2.2.3. Strain Displacement

The strain displacement is expressed by
(11)$$\frac{1}{2}\left[\vec{e_{i}}\frac{\partial \vec{D}}{\partial x_{i}}+(\vec{e_{i}}\frac{\partial \vec{D}}{\partial x_{i}}{)}^{T}\right]=\varepsilon ,$$
where ε with three normal strains and six shear strains is the strain tensor (m), which is the same as the stress tensor.

#### 2.2.4. Elasticity

The elasticity is described by
(12)$${p=2G\varepsilon +K}_{s}tr\left(\varepsilon \right)I,$$
where *I* is the unit tensor, *G* denotes the modulus of elasticity, and $K_{s}$ expresses the thermal conductivity of the refractory material.

Combining Equations (10) and (11) with Equation (12), the governing equation of the elastic body is expressed by the displacement.
(13)$$\frac{\partial^{2}\left(\rho_{s}D\right)}{\partial t^{2}}-\vec{e_{i}}\frac{\partial }{\partial x_{i}}\cdot [G\vec{e_{i}}\frac{\partial \vec{D}}{\partial x_{i}}+G{\left(\vec{e_{i}}\frac{\partial \vec{D}}{\partial x_{i}}\right)}^{T}{+K}_{s}Itr\left(\vec{e_{i}}\frac{\partial \vec{D}}{\partial x_{i}}\right)]=0,$$

Adding the influence of temperature on the displacement to the elastic body described by Equation (10), the stress equation affected by temperature and the displacement is obtained from
(14)$$\frac{\partial^{2}\left(\rho_{s}D\right)}{\partial t^{2}}-\vec{e_{i}}\frac{\partial }{\partial x_{i}}\cdot \left[G\vec{e_{i}}\frac{\partial \vec{D}}{\partial x_{i}}+G{\left(\vec{e_{i}}\frac{\partial \vec{D}}{\partial x_{i}}\right)}^{T}{+K}_{s}Itr\left(\vec{e_{i}}\frac{\partial \vec{D}}{\partial x_{i}}\right)\right]+\frac{\partial \left({aT}_{s}\right)}{\partial t}=0,$$
where a is the thermal expansion coefficient of the refractory material.

In order to improve the convergence, the equation is written in a new form
(15)$$\frac{\partial^{2}\left(\rho_{s}D\right)}{\partial t^{2}}-\vec{e_{i}}\frac{\partial }{\partial x_{i}}\cdot \left[{(2\mu +K}_{s})\vec{e_{i}}\frac{\partial \vec{D}}{\partial x_{i}}\right]+\vec{\;e_{i}}\frac{\partial }{\partial x_{i}}\cdot \left[G{\left(\vec{e_{i}}\frac{\partial \vec{D}}{\partial x_{i}}\right)}^{T}{+K}_{s}Itr\left(\vec{e_{i}}\frac{\partial \vec{D}}{\partial x_{i}}\right)-{\;(G+K}_{s})\vec{e_{i}}\frac{\partial \vec{D}}{\partial x_{i}}\right]+\frac{\partial \left({aT}_{s}\right)}{\partial t}=0,$$

### 2.3. Conjugated Heat Transfer

Refractory materials and hot metal exchange energy, and the increased in heat in one phase is equal to the decrease in heat in the other phase. Therefore, the heat transfer formula of the solid–liquid interface is
(16)$$K_{f}\frac{{dT}_{f}}{d\vec{n}}=-K_{s}\frac{{dT}_{s}}{d\vec{n}},$$
where $\vec{n}$ represents the normal direction on the solid–liquid interface.

## 3. Model Configuration

### 3.1. Mathematical Model

The geometry of the mathematical model in the simulations is the same as that of the trough of No. 2 BF in Lai Steel, as shown in Figure 1 [19]. During tapping, hot metal flows out of the taphole with an angle of 10°, and then falls into the main trough. The height of hot metal from the upper surface of the main trough is 300 mm. The dropping position of hot metal is defined as the inlet of the main trough (the initial position is 4 m away from the taphole). As the tapping progresses, the speed of inlet metal flow gradually decreases, the inlet position moves toward the taphole as the tapping progresses. The refractory of working layer is the most severely eroded part of the main trough; therefore, only this part is studied in this paper, and the thickness of the refractory material on both sides is 550 mm. Based on previous studies [6,10] and plant data, the physical properties of the fluid and the refractory in the study are as presented in Table 1. The main component of the refractory castable is Al_2_O_3_-SiC-C and its thickness is shown in Figure 1b.

The computational grid of the main trough model used in this article, as shown in Figure 2, has been proven to give mesh-independent results for an average mesh size of 20 mm or smaller. Therefore, the model with 20 mm grids was used in the simulations. This model contains 6433548 hexahedral cells. The mesh size was maintained as 20 mm for the different cases even though the trough geometry was changed in some of the cases studied.

### 3.2. Assumptions

The following assumptions are made to simplify the model:(1)Hot metal is an incompressible Newtonian fluids;(2)Thermophysical parameters are constant;(3)The Marangoni effect and chemical reactions are ignorable;(4)Only hot metal is considered in the main trough;(5)The main trough lining is intact and experiences no corrosion.

### 3.3. Boundary Conditions

#### 3.3.1. Inlet Boundary Conditions

The injection process of hot metal from the taphole to the dropping position is regarded as an oblique throwing motion [5] described by
(17)$$y=x\cdot tan\alpha -\frac{g}{{2u}_{0}^{2}cos^{2}\alpha }{\cdot x}^{2}$$
where *α* denotes the taphole angle, $u_{0}$ is the metal flow velocity at the taphole, while *x* and *y* are the coordinates of the dropping position of hot metal.

The tapping parameters at the reference plant are given in Table 2 and Figure 3. The taphole angle, the location of the taphole, the falling height of the hot metal and the position of the farthest drop point are used in Equation (17), and the maximum tapping speed is calculated. Combining the maximum tapping speed at the mouth with the Equation (17), the hot metal velocity at the entrance to the upper surface of the hot metal is obtained. As the flow rate at the inlet of the falling hot metal is equal to the one at the taphole, the entrance size is realized. The inlet temperature is 1773 K and the pressure gradient is zero.

#### 3.3.2. Outlet Boundary Conditions

An outlet pressure of p=100,000 Pa (1 atm) is adopted as the boundary condition. Velocity, temperature, turbulent kinetic energy and turbulent energy dissipation rate are all set to have no gradients.

#### 3.3.3. Wall Boundary Conditions

The wall surface of the refractory material satisfies the non-slip condition, so the wall surface applies the wall function boundary conditions to define the flow near the wall surface. The energy transfer between the refractory material and the main trough is described by a conjugate heat transfer expression [20]. The heat transfer coefficient of natural convection between the upper surface of the main trough and the air is 10 W × (K·m^2^)^−1^ [21]. The initial temperature of the hot metal in the main trough is 1723 K. The initial temperature of the ramming material of the main trough working layer and the outer refractory material is set to 1273 K.

#### 3.3.4. Internal Field Boundary Conditions

The initial value is the same as the initial value at the entrance and the pressure gradient is set to zero.

#### 3.3.5. Time Step and Sub-Iterations

The time step size of the calculation is set to 0.001 s. In order to ensure the achievement of steady state, the number of sub-iterations was set to 10,000. Tapping time of No. 2 BF in Lai Steel is 90 min. A 90-min simulation may lead to a waste of computing resources, hence the time of hot metal tapping in the model was set to 90 s. The calculation time is about 90 s in the case where the accuracy of the calculation results is guaranteed.

## 4. Results and Discussion

### 4.1. Velocity Distribution of the Fluid in the Main Trough

The inspection sections and lines in the main trough are presented for the overall description of physical fields in Figure 4.

Figure 5 shows the velocity distribution in the central section of the main trough along the metal flow direction (cf. Figure 4). In the early stage of tapping, the drop point is located 3.5 m from the left side of the main trough (the blue point). At this moment, there is obvious turbulence downstream, and the turbulence region falls within the range of 4.5~6 m. In the middle of hot metal tapping, the hot metal drop point moves towards the taphole. At this moment, turbulence still exists, and the range of action is expanded to 3.5~6.5 m. It is due to the increase in metal flow angle at the drop point of the hot metal (37°–50°). Therefore, the direction angle of metal flow becomes larger after metal flow is blocked by the bottom of the main trough, and turbulence is more likely to occur. Although the flow velocity of the falling metal is reduced, turbulence is fully developed. As tapping continues, the drop point moves further towards the taphole and the flow velocity of the falling metal flow decreases. However, the increase in the angle of the drop point contributes to a more intense scour effect of the hot metal flow on the bottom of the main trough. The shape of the turbulence region becomes narrower and longer, with an influence range of 2~6.5 m.

### 4.2. Temperature Distribution of the Main Trough

Figure 6 shows the temperature distribution of the central section of the main trough at different time points. Three lines are presented for the overall description of temperature. As the tapping progresses, the temperature increases significantly and the temperature changes greatly in the early and interim tapping stages. For Line 1, the extreme temperature of the initial time is increased by 11 K higher than the end time. For Line 2, the extreme temperature of the initial time is increased 10 K higher than the end time. During the interim to final tapping, the temperature experiences a small change, indicating that the temperature in the main trough tends to stabilize after 45 min of tapping at about 1740 K. Han et al. [22] found that the hot metal temperature gradually stabilized after 40 min of tapping in the tapping trough, so the findings agree well.

The temperature change curves of three different tapping times are drawn along the bottom and side of the main trough refractory material as shown in Figure 7. The temperature of the main trough sidewall and the main trough working layer increase as the hot metal is tapped. The heat preservation of the refractory material is good and thermal conductivity of it is much less than that of hot metal, so the temperature rise is small with a change range of about 17 K. At the end of tapping, the temperature shows the highest value. The center of the sidewall and the center of the bottom of the main trough has a better heat transfer where the temperature change is the largest, and the temperature change in the edge area is minor.

Thermocouples are installed in the main trough to record temperature values every 30 min and the tapping time of a BF is about 90 min. Therefore, there are about four temperature records for a tapping. The calculation time is 90 s in our cases. Since there is a fluctuation of the monitoring temperature, an averaging temperature of the four-time segments in hot metal is necessary. Comparisons between the simulated and measured temperatures in the plant are shown in Figure 8. During the tapping, the temperature of on-site monitoring and temperature of the numerical calculation fluctuate with a similar trend and the latter changes in a small range. The average temperature error is 1.3% so the temperature distributions agree reasonably well. Thus, the simulations are consistent with plant measurements.

### 4.3. Wall Shear Stress

Figure 9 shows the wall shear stress distribution of the main trough at the initial, middle and final stages of the tapping, respectively, and the shear force distribution on the straight line is obtained. The shear stress on the bottom surface of the main trough is seen to be greater than on the sidewall, while the shear stress near the hot metal drop point is greater than at other locations. During the tapping process, the shear stress at the upper part of the sidewall gradually decreases, and the shear stress concentration area is found between 4 m and 5 m. The shear stress between the lower part of the sidewall and the bottom surface gradually increases, and the shear stress concentration area of the two is approximately the same, located at 3~7 m.

### 4.4. Wall Thermal Axial Stress

The equivalent stress of the main trough as a whole is shown in Figure 10, which demonstrates that the location of the stress concentration is related to the location of the falling metal flow. Due to the injection of high-temperature hot metal, the temperature gradient increases rapidly in the early stage of the tapping. The average stress value of the refractory material downstream of the main trough is less than at other tapping times, and the stress at the entrance of the iron dam and the iron trough is also greater. In the middle of casting, there is no turbulence, and the temperature gradient is reduced, so the stress value is reduced compared with the initial stage of the tapping. Therefore, the stress near the hot metal drop point and downstream of it still remains high. At the end of tapping, due to the increase in the angle at the hot metal drop point, the falling hot metal does not flow directly downstream when it reaches the bottom surface of the trough, but instead flows to the sidewall. Under the effect of the high-temperature hot metal, the temperature gradient at the lower part of the sidewall and the stress value increase. At the same time, due to the edges and corners of the lower part of the sidewall, the stress concentration is aggravated, so the thermal stress in the lower part of the sidewall becomes larger.

The stress value near the hot metal drop point is larger than at other positions. Three cross-sections at 3.5 m, 4 m and 4.5 m were selected for analysis and the results are shown in Figure 11. The temperature gradient on the surface of the lining is large, forming a thin stress-concentration layer. The temperature gradient in the early stage of the tapping is relatively large, so the range of the high-stress distribution is larger than that in the middle of the tapping. However, as the tapping continues, the turbulence in the main trough changes, the temperature gradient in the refractory material increases, and the stress-affected area expands again.

Eight straight inspection lines were selected on the bottom of the main trough, as depicted in Figure 12, to study the stress distribution. It can be seen from Figure 13 that from the center to the sidewall, the thermal stress at different positions is not much different. In the early stage of the tapping, the thermal stress downstream is very large and the stress upstream and downstream in the main trough are significantly different. In the middle of the tapping, the thermal stress value decreases significantly, because of the relatively stable temperature and gentle rise of the hot metal temperature at this time. The temperature gradient is also much smaller than at the initial stage of the tapping. At the end of the tapping, the stress value increases again, but the peak value is smaller than at the beginning of the tapping.

Based on the analysis of the shear force and thermal stress, it is concluded that the refractory material on the bottom of the main trough experiences greater thermal stress during the tapping process, and the maximum thermal stress occurs at 4 m on the bottom of the main trough. Therefore, key points for further analysis are selected at this position and in nearby areas in order to observe the stress changes in a working cycle. After calculating the tapping process (which lasted 90 s in the simulations), an equally long period (“tapping gap”) without hot metal supply was simulated to study the working state of the main trough during the tap cycle. This gave the stress curve depicted in Figure 13.

Since R ≠−1 (R is the stress ratio) for the stress curve, an average stress correction is required. The Goodman average stress correction method was applied, using the formulas
(18)$$S_{a}=\frac{S_{max}-S_{min}}{2}$$
(19)$$S_{m}=\frac{S_{max}{+S}_{min}}{2}$$
(20)$$\frac{S_{a}}{S_{a(-1)}}+\frac{S_{m}}{S_{u}}=1$$
where $S_{\text{max}}$, $S_{min}$ and $S_{a-1)}$ represent the maximum stress, the minimum stress and the corrected stress amplitude (Pa), respectively. $S_{u}=1{.92\;\times \;10}^{7}\;\mathrm{Pa}$ is the ultimate tensile strength of refractory material [23].

Substituting the corrected result into the S-N stress fatigue life curve, shown in Figure 14 [23], new results are obtained: the peak stress at 4 m is $1{.68\;\times \;10}^{7}$ Pa, fatigue life is about: 190 cycles of loading (the number of tapping times is 190). The fatigue life at 3.5 m, 3 m and 2.5 m are about 260 times, 290 times and 340 times, respectively. A single main trough of Laiwu Steel’s No. 2 BF taps hot metal seven times a day on average and is repaired once every 25 days. The main trough was thus tapped 175 times during the real operation, which agrees well with the results of the model.

### 4.5. Effect of Baffle Size

An optimization of the main trough structure can be undertaken to suppress the formation of turbulence and reduce the impact of backflow. Suppressing the formation of turbulence in the main trough is here the top priority. On the basis of Section 4.1, it can be seen that the counter-clockwise turbulence in the main trough recirculates between 5.8 m and 6.2 m, so baffles were set at this position in the model, as reported in Table 3.

#### 4.5.1. Main Channel Flow Field

When the tapping has lasted 30 min, the simulated velocity distribution of each cross-section of the main trough is presented in Figure 15.

The figure shows that the installation of baffles changes the distribution of the flow field. At the 5 m cross-section, the area is affected by the return flow on the upper side of the main trough that becomes larger because of the downstream baffles. At the 5.5 m and 5.7 m sections, the velocity concentration area in the lower part of the trough is far from the bottom. On the sidewall, due to the blocking of the baffles, more hot metal flows downstream and the intensity of the turbulence is weakened. At the 6 m position, the hot metal at Baffle 1 and Baffle 2 is affected by the baffles, which causes the lateral width to decrease, and the flow cannot be fully diffused. Therefore, more hot metal flows from the middle of the baffles downstream without obvious speed concentration.

Figure 16a presents the velocity distribution near the sidewall (Line 5) at 5.7 m in the metal flow direction of the main trough. Due to the influence of baffles, the extreme speed on the upper sidewall for the Baffle 1 and Baffle 2 cases is decreased by 4.8% and 9.2%, respectively, compared with the base model, which shows that adding baffles here can suppress turbulence, and the turbulence speed on the sidewall is significantly reduced. In order to verify this conclusion, the conditions along Line 1 on the upper side of the main trough is studied. Figure 16b shows that the extreme speed of the Baffle 1 case is reduced by 4.8% and for the Baffle 2 case by 4.6%. The velocity distribution at the sidewall is shown in Figure 16c. Affected by turbulence, there are more peaks at the sidewall, but the extreme velocity of each model has decreased. The middle channel flows away from the lower part of the sidewall. From Figure 16d, showing the velocity distribution on the bottom surface of the main trough, it can be seen that the speed at the bottom surface has dropped significantly, especially for the case with Baffle 2, and is decreased by 6.8% compared with the base model.

The flow at the two cross-sections 5.5 m upstream and 6.6 m downstream are selected for analysis, and the results are shown in Figure 17. The volume flow of the Baffle 1 and Baffle 2 models has decreased compared with the base model: the volume flow of Baffle 1 has dropped by 5.0% and for Baffle 2 by 6.4%. In Figure 17b, the addition of baffles effectively increases the downstream flow rate. It is due to the reduction in the return metal flow rate and direct downstream of much more hot metal, resulting in an increase in the downstream surface flow rate by 11%.

#### 4.5.2. Stress analysis

Line 1 on the sidewall of the main trough and Line 2 at the bottom of the main trough are analyzed as shown in Figure 18. The shear stress of Baffle 2 is the smallest, and the decrease is about 26%. The addition of baffles is seen to reduce the time-averaged shear stress on the main trough wall, the hot metal to the main trough and the erosion of the sidewall. However, whether the addition of baffles reduce the time-averaged shear stress at the bottom of the main trough is unknown.

#### 4.5.3. Fatigue Life

Results of an analysis of the stress fatigue of the bottom surface of the main trough are shown in Figure 19. The stress values at the three locations on the bottom of the trough show little change. The minimum fatigue life of the Baffle 2 model at 4 m is estimated to be increased by 15 tappings (about 2.5 days of operation) from the S–N curve, which is shown in Figure 14.

### 4.6. Effect of Baffle Width and Length

The second research point in this study lies in the inhibition backflow. In this section, baffles are set at the position where the recirculation is strong, aiming to suppress the backflow and reduce the erosion on the sidewall. The study is divided into two parts. The influence of baffle width on the suppression of reflux is studied by the Baffle 3, 4 and 5 models, while the influence of baffle height is analyzed by the Baffle 5, 6 and 7 models. Detailed parameters of these setups are shown in Table 4.

#### 4.6.1. Flow Field Analysis

The velocity distribution of each cross-section of the main trough is presented in Figure 20 for the case where hot metal has been tapped for 30 min. Figure 20a shows that the velocity distribution of each model in the 3.5 m cross-section is quite different. As the baffles are located at 4~5 m in the main trough block the return metal flow, only a small amount of metal flow reaches the side wall of the main trough before the 4 m location. The figures present a general decline in velocity as the width of the baffles (in the z direction) increases, as the blocking effect becomes more significant. It can be seen from the 3.5 m cross-section of Figure 20b that as the height of the baffles increases, the metal flow rate upstream of the baffles is significantly reduced. Baffle 7 with the lowest height does not inhibit the backflow very well, so return hot metal flows from the upper end of the baffles.

In order to analyze the change of the metal flow velocity at the wall, the conditions along two straight lines, blue (120 mm from the sidewall) and white (20 mm from the baffle’s wall) are selected for the analysis near the wall of each model. The results are shown in Figure 21.

Figure 21a indicates that the extreme velocity of the base model is the largest, while the velocity of the baffle model is reduced. The width of Baffle 3 is the largest and the velocity drop rate is largest, 12.2%. The results along a straight line 20 mm from the wall in all models are shown in Figure 21b: at the same distance from the wall, the metal flow velocity of the baffle models is less than that of the base model. The extreme velocity of Baffle 5 is 0.233 m/s, and the extreme velocity of the base model is 0.246 m/s, so a decrease of about 5% is obtained by the design.

Figure 22a depicts the velocity distribution along the white line on the sidewall of the main trough (20 mm from the baffle’s wall). The extreme velocity of Baffle 7 shows the largest decrease 11.7%, because of the different analysis positions. The recirculation velocity near the upper surface is overall larger, while the metal flow velocity near the upper surface is smaller. After analyzing the results along Line 4 on the cross-section of the main trough, the result shown in Figure 22b was obtained. At the center of the main trough, the metal flow velocity basically does not change. At the edge of the trough, the speeds of the Baffle 5 and Baffle 7 cases are significantly lower than those of the base model; the largest decrease is about 10%. Baffle 6 is a too low design to prevent hot metal from flowing near the wall. Figure 22c shows the velocity distribution along Line 5 near the sidewall of the trough. It is seen that the velocity of the model with a baffle near the sidewall is less than in the base model, the largest decrease about 33%. Due to the blocking effect of the baffle, the metal flow velocity near the upper and lower sides is lower than in the base model. In the vicinity of the baffle wall, since the baffles block the backflow, the speed of the hot metal flow is greater than in the base model.

Since baffles are set at a position of 4~5 m from the taphole, a cross-section at a distance of 4.5 m from the taphole is selected to study on the effect of baffles on reflux suppression. For main troughs of different widths, as shown in Figure 23a, the flow rate of the Baffle 3 case is the smallest among the four models. Compared with the cross-sectional flow of the base model, the flow rate drops by about 3.2%, and the flow rate of the Baffle 5 model drops by about 3.1%. The decrease in flow rate is inversely proportional to the width of the baffles, that is, the larger the width, the better the suppression effect on the backflow in the main trough. Figure 23b shows the flow rate of each model cross-section downstream of the baffles. Since the reflux metal flow is blocked by the baffles, the flow rate of the downstream cross-section increases, and the flow rate is inversely proportional to the upstream flow rate. It is shown that the hot metal flows downstream after being blocked by the baffles. For main troughs of different heights, the cross-sectional flow of the main trough at 4.5 m (the middle position of baffles) and 5.2 m (downstream of baffles) was calculated. The results shown in Figure 24 indicate that as the height of the baffles increases, the flow rate at the 4.5 m section increases significantly.

#### 4.6.2. Stress Analysis

The influence of the addition of baffles on the wall was studied and the time-averaged shear stress of the main trough wall was analyzed as shown in Figure 25 and Figure 26.

Figure 25a shows that since the baffles added in this part are on the sidewalls of the trough, they have little effect on the metal flow at the bottom of it. Therefore, all models with baffles added decrease the shear stress slightly compared with the base model. Figure 25b reveals that the maximum time-averaged shear stress of the model with baffles is greater than that of the base model. It is inconsistent with the analysis result of the flow field. The metal flow near the wall decreases because the upstream flow rate decreases, and the metal flow velocity slows down. The increase in wall shear occurs as backflow is partially obstructed by the baffles. It can be seen from Figure 26 that the shear stress on the wall of the model with baffles has been reduced, and the effects of Baffle 5 and Baffle 6 are better. As the position of Baffle 6 is relatively high, the effect of restraining the flow in this part is better. The shear stress on the wall of the baffles is greater than for other models, but the position of Baffle 7 is too low, and the backflow appears at the top of the baffles, leading to greater shear stress. 

For baffles of different widths, the inhibitory effect of Baffle 5 on the reflow is significantly different from the case with baffles of maximum width, and the shear stress for Baffle 5 is smaller than for Baffle 3. As for the study of baffles of different heights, Baffle 7 has the best effect on reducing the reflow on the sidewall, but it is not as good as Baffle 6 and Baffle 5 in suppressing the time-averaged shear stress on the sidewall. Comprehensively, Baffle 5 can not only effectively reduce the upstream flow, but also concentrates most of the shear force on the baffles, which is the best solution to reduce the detrimental effect of backflow on the sidewall.

## 5. Conclusions

The main trough of the blast furnace is the only passage for hot metal and slag leaving the taphole for the torpedo and the slag granulation unit, and an important place to separate slag from hot metal and desulfurize hot metal as well. Studying the working status of the main trough and the erosion mechanisms of the refractory can provide useful information that can be used to improve the life length of the main trough lining and save costs of hot metal production. A mathematical model based on OpenFOAM was applied to analyze the flow temperature, wall shear stress, wall thermal stress in the main trough and the influence of adding baffles. The main conclusions are as follows:(1)Turbulence intensity downstream of the hot metal dropping position becomes weaker and the turbulence ranges become larger (2~6.5 m from the beginning of the trough);(2)Maximum thermal stress appears at 4 m from the beginning of the main trough, which is the position of the minimum fatigue life of the trough. In the simulation, fatigue life of a new trough refractory is estimated to be 190 times hot metal’s tapping, which agrees well with the practice in the steel plant;(3)Installation of baffles of the sidewall at the 5.8~6.2 m position can suppress the turbulence of hot metal. The suppression effect of a big baffle (named Baffle 2 in this study) is better than a small one (Baffle 1), and the minimum fatigue life of the main trough is estimated to increase by 15 tappings (i.e., about 2 days of operation);(4)Setting up baffles at the upper part of the sidewall at the 4~5 m position can inhibit the backflow of hot metal. Among the studied alternatives, Baffle 5 shows the highest inhibition performance on the backflow, but the minimum fatigue life remains unaffected.

## Figures and Tables

**Figure 1 materials-14-04851-f001:**
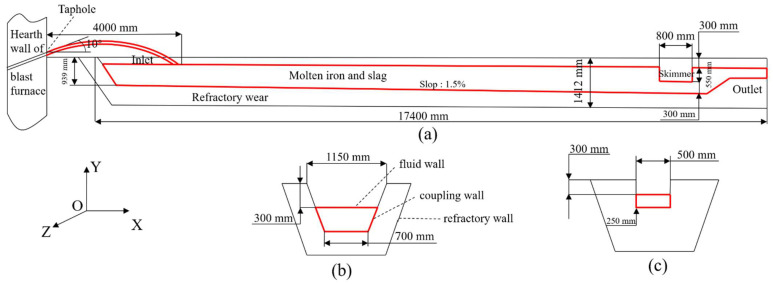
System studied and its dimensions in the simulation: (**a**) front view, (**b**) cross-section at inlet, (**c**) cross-section at outlet.

**Figure 2 materials-14-04851-f002:**
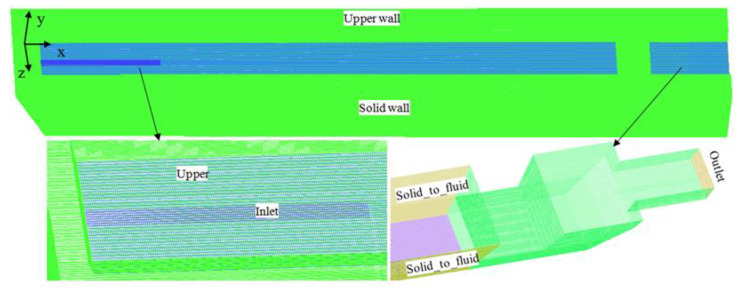
Computational grids of the main trough.

**Figure 3 materials-14-04851-f003:**
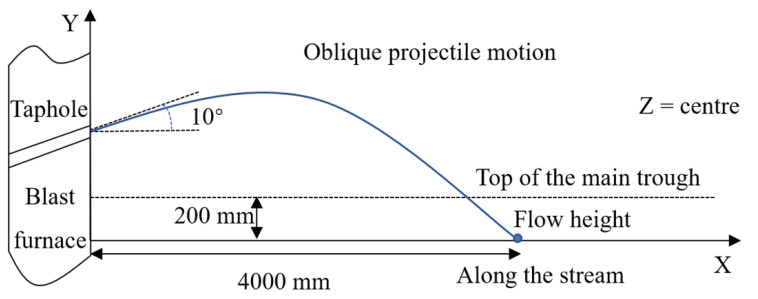
Dropping position calculation of the hot metal.

**Figure 4 materials-14-04851-f004:**
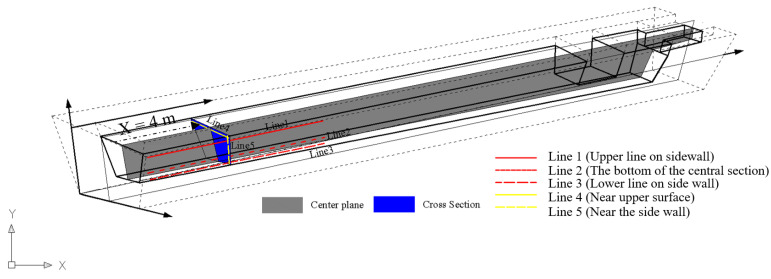
Post-processing inspection positions for transient calculation of the main trough.

**Figure 5 materials-14-04851-f005:**
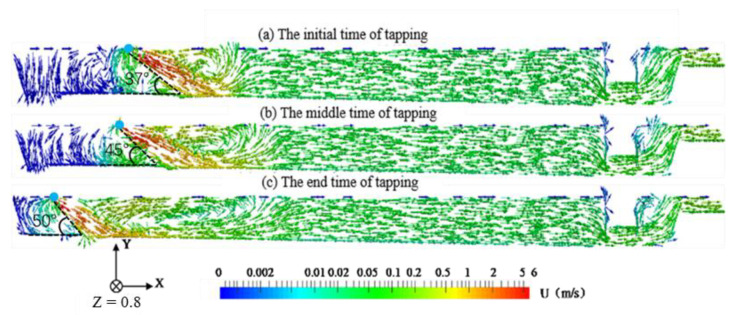
Velocity distribution of hot metal on the center plane (cf. Figure 4) at three different time points.

**Figure 6 materials-14-04851-f006:**
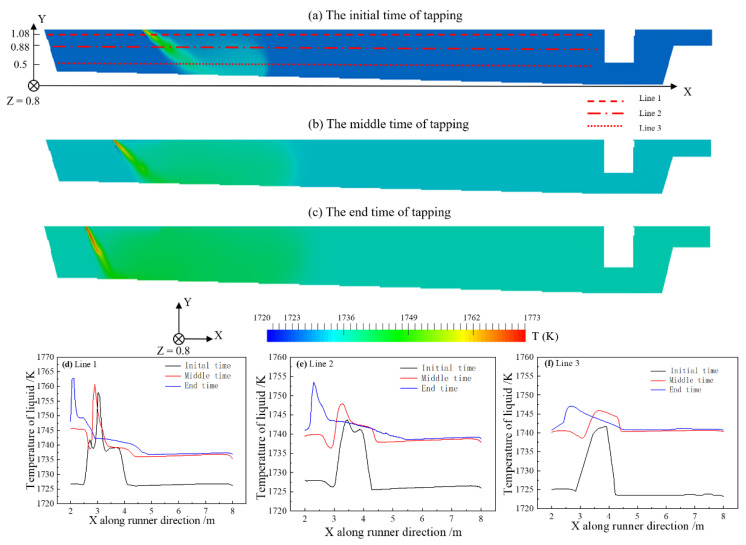
Temperature distribution of hot metal on the center plane (cf. Figure 4) at three different times: (**a**) Initial; (**b**) Middle; (**c**) End and different Y-axis height: (**d**) 1.08 m; (**e**) 0.88 m; (f) 0.5 m.

**Figure 7 materials-14-04851-f007:**
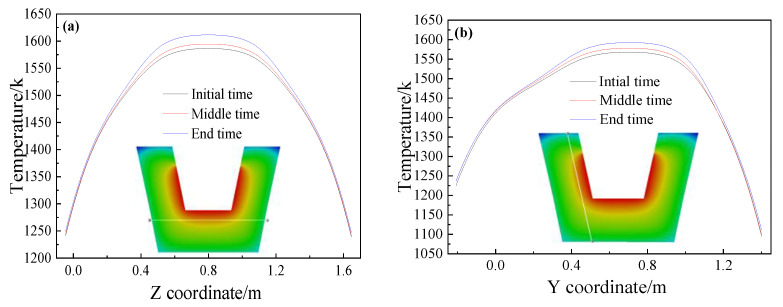
Refractory temperature of the main trough at different positions during tapping: (**a**) X = 4 m, Y = 0.35 m; (**b**) X = 4 m, Z = 0.4 m.

**Figure 8 materials-14-04851-f008:**
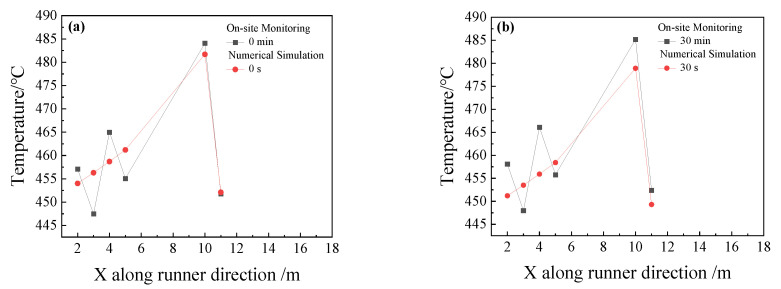
Temperature comparisons in the simulations and in plant measurements at four different times: (**a**) 0 s,0 min; (**b**) 30 s,30 min; (**c**) 60 s,60 min; and (**d**) 90 s,90 min.

**Figure 9 materials-14-04851-f009:**
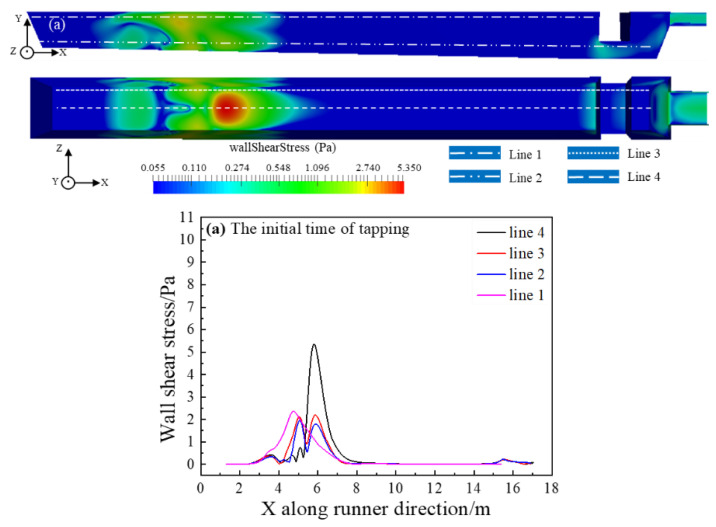
Wall shear stress distribution at three different times: (**a**) Beginning; (**b**) Middle; and (**c**) End.

**Figure 10 materials-14-04851-f010:**
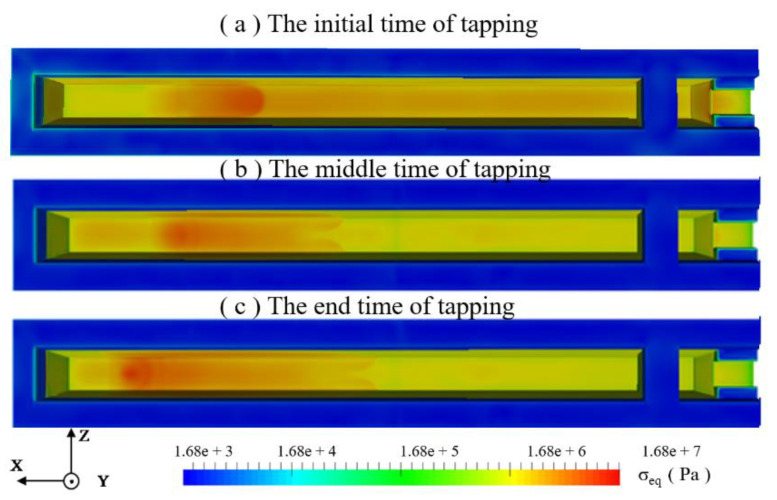
Equivalent stress distribution of the main trough at different times.

**Figure 11 materials-14-04851-f011:**
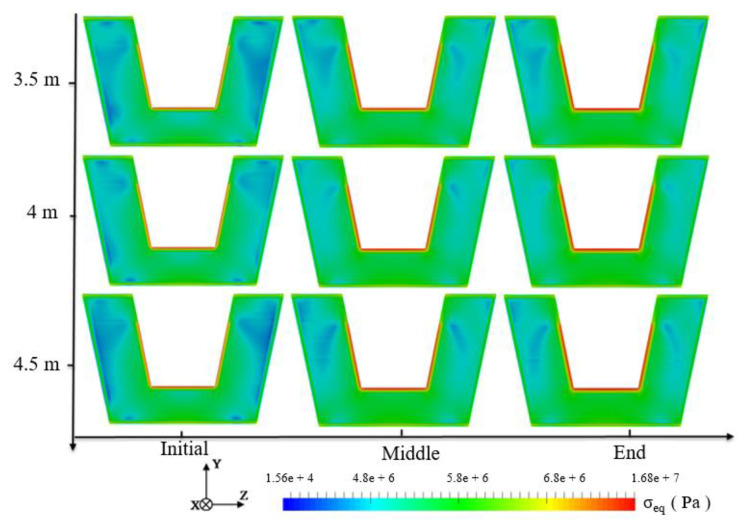
Equivalent stress distribution on the cross-section of the main trough at different times.

**Figure 12 materials-14-04851-f012:**
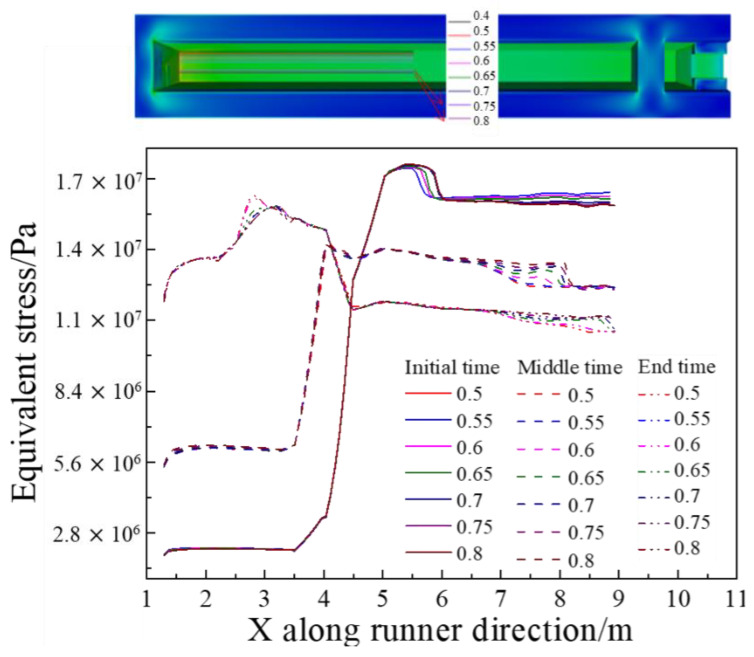
Equivalent stress fluctuations of the bottom of the main trough at different times.

**Figure 13 materials-14-04851-f013:**
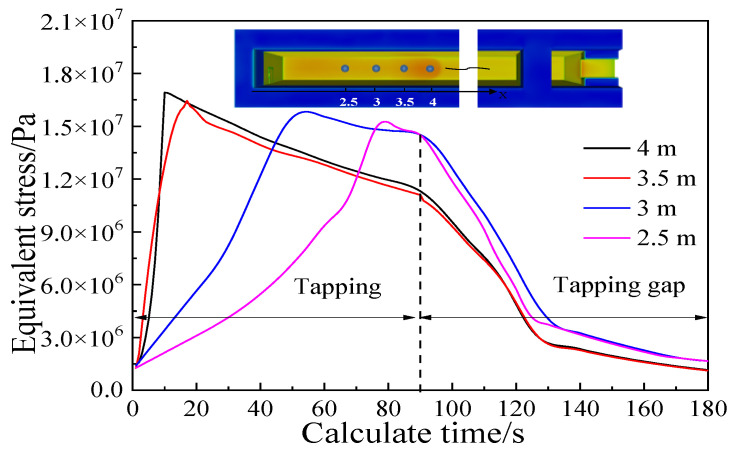
Stress curve of key points during tapping.

**Figure 14 materials-14-04851-f014:**
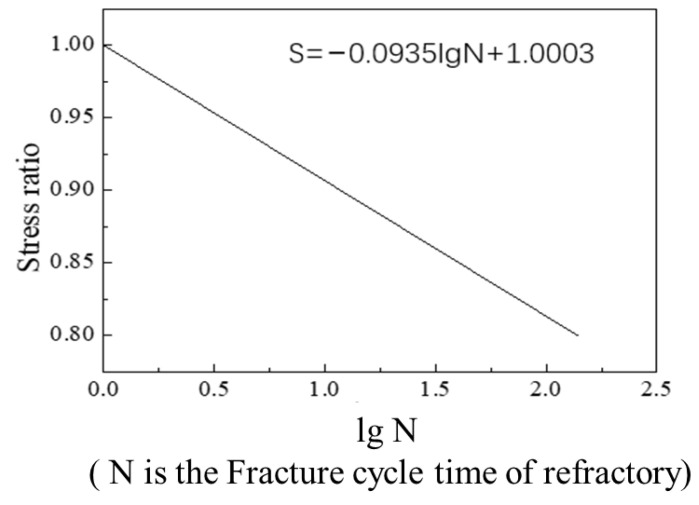
Low cycle fatigue characteristic curve of the working layer [23].

**Figure 15 materials-14-04851-f015:**
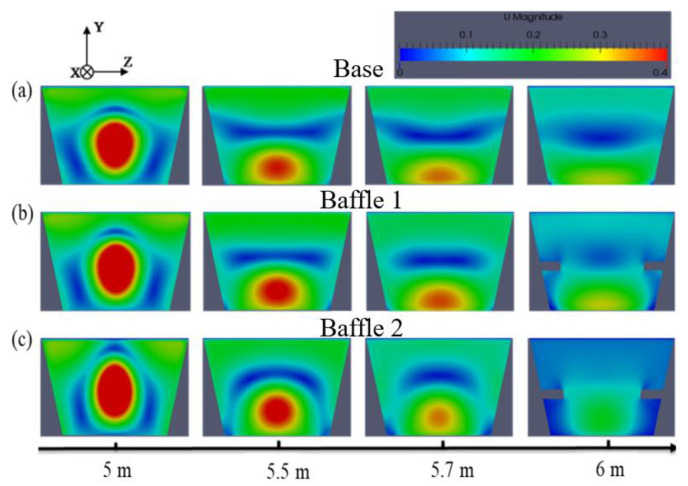
Effect of adding baffles that suppress turbulence on the velocity distribution of the main trough: (**a**) Base; (**b**) Baffle 1; and (**c**) Baffle 2.

**Figure 16 materials-14-04851-f016:**
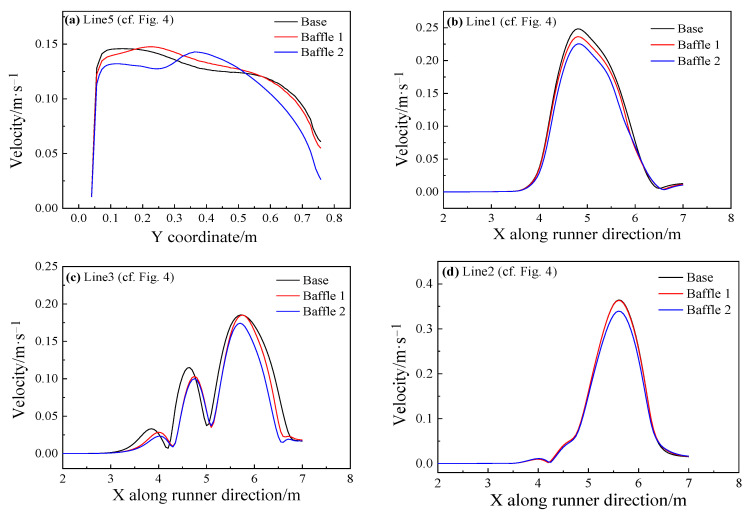
Velocity distribution of different main troughs near the wall without or with baffles.

**Figure 17 materials-14-04851-f017:**
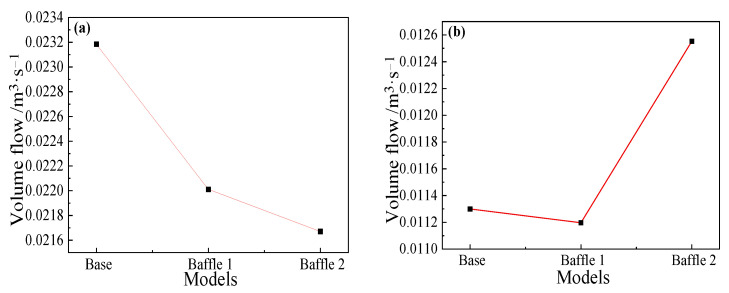
Volume flow at positions (**a**) upstream (5.5 m) and (**b**) downstream (6.6 m) of the baffle for different main troughs.

**Figure 18 materials-14-04851-f018:**
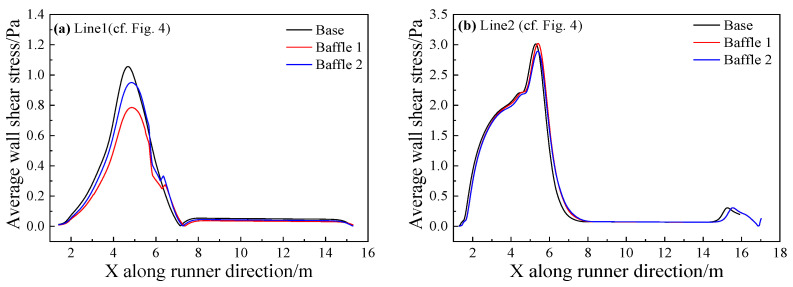
Time-averaged wall shear stress of the different main troughs.

**Figure 19 materials-14-04851-f019:**
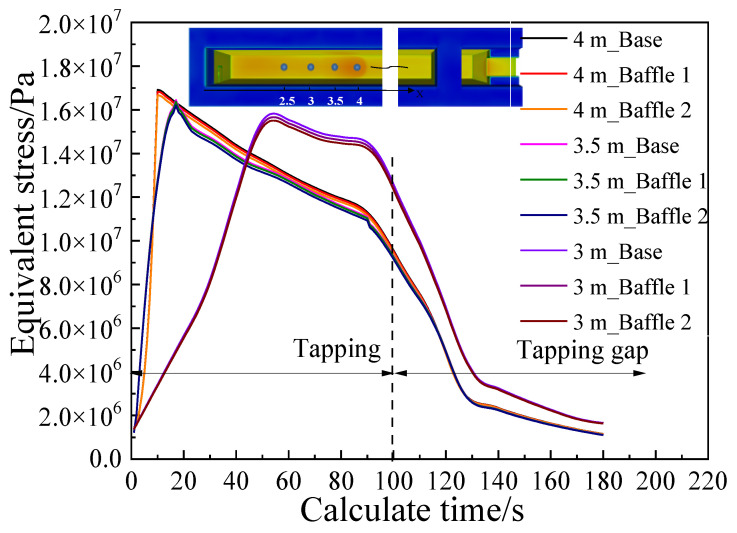
Stress load on the bottom of main troughs with different widths and depths.

**Figure 20 materials-14-04851-f020:**
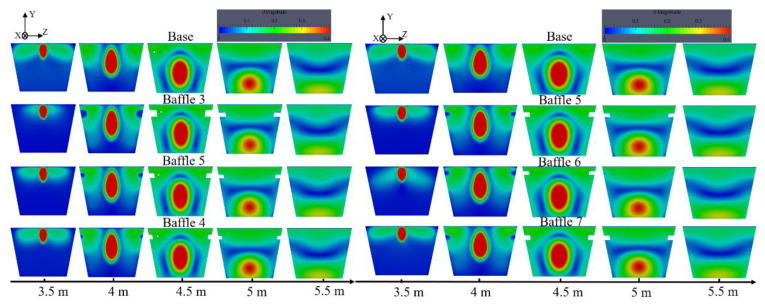
Effect of adding baffles with different widths (**a**) and different heights (**b**) to suppress the reflux on the velocity distribution of the main trough.

**Figure 21 materials-14-04851-f021:**
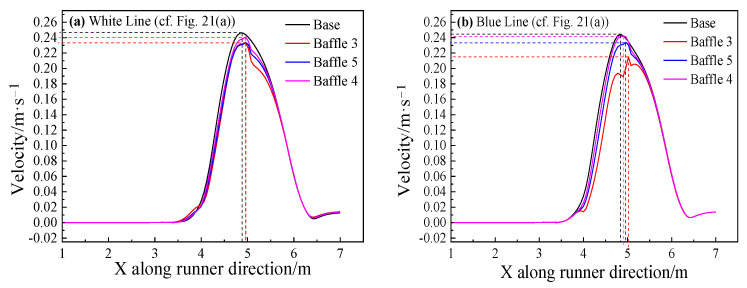
Velocity distribution of different main troughs with different widths near the wall at two lines: (**a**) White Line; (**b**) Blue Line.

**Figure 22 materials-14-04851-f022:**
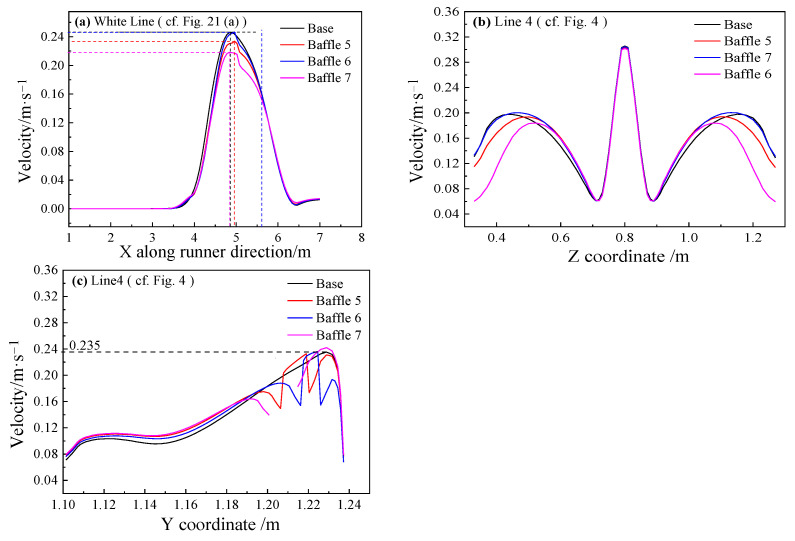
Velocity distribution of different main troughs with different heights near the wall at different lines: (**a**) White Line; (**b**) Line 4; (**c**) Line 4.

**Figure 23 materials-14-04851-f023:**
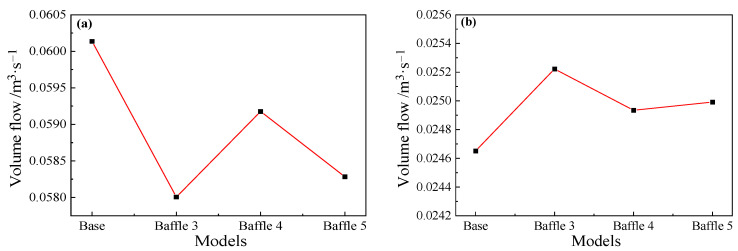
Volume flow at positions: (**a**) upstream (4.5 m); and (**b**) downstream (5.2 m) of the baffle for main troughs with different widths.

**Figure 24 materials-14-04851-f024:**
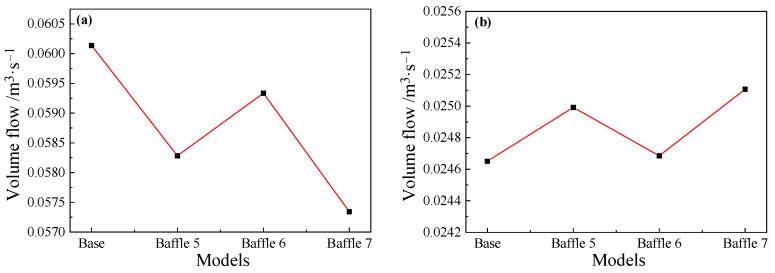
Volume flow at positions: (**a**) upstream (4.5 m); and (**b**) downstream (5.2 m) of the baffle for main troughs with different heights.

**Figure 25 materials-14-04851-f025:**
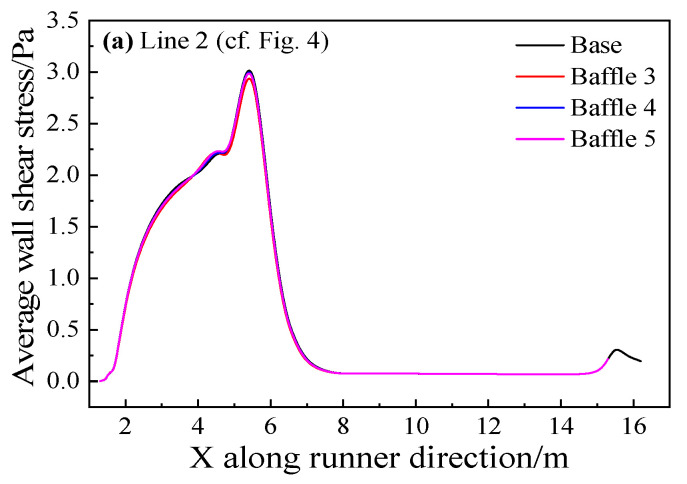
Time-averaged wall shear stress of the different main troughs at two lines: (**a**) Line 2; and (**b**) Line 1.

**Figure 26 materials-14-04851-f026:**
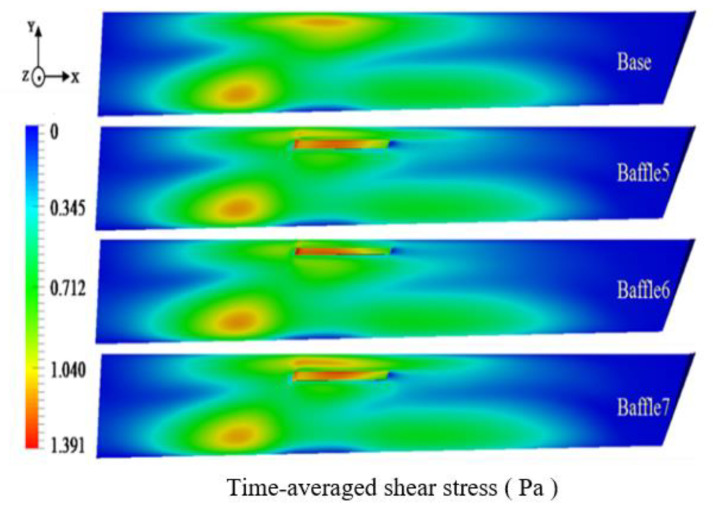
Time-averaged shear stress distribution on the sidewall of the main trough with different heights.

**Table 1 materials-14-04851-t001:** Physical parameters of the main trough.

Property	Value
Density of hot metal (kg × m^−3^)	6900
Viscosity of hot metal (kg × m^−1^×s^−1^)	0.0045
Thermal conductivity of hot metal (W × m^−1^ × K^−1^)	16.5
Specific heat capacity of hot metal (J × kg^−1^ × K^−1^)	850
Hot metal production rate (t × d^−1^)	3400
Density of the refractory (kg × m^−3^)	2850
Thermal conductivity of the refractory (W × m^−1^ × K^−1^)	2.6
Specific heat capacity of the refractory (J × kg^−1^ × K^−1^)	750

**Table 2 materials-14-04851-t002:** Tapping parameters of No. 2 BF in Lai Steel.

Parameters	Value
Diameter of taphole	60 mm
Angle of taphole	10°
The farthest dropping position of molten iron (DPMI)	4 m
Hot metal level in main trough	200 mm
Tapping time	90 min
Number of tappings per day	14

**Table 3 materials-14-04851-t003:** Size of the baffles that suppress turbulence.

Baffle Size	Baffle 1 (Small)	Baffle 2 (Big)
Length (X)/mm	500	800
Width (Z)/mm	165	165
Height (Y)/mm	100	100

Baffle 1 position: X (5800 mm from the taphole), Y (the midpoint of the main trough wall), Z (the wall). Baffle 2 position: X (5635 mm from the taphole), Y (the midpoint of the main trough wall), Z (the wall).

**Table 4 materials-14-04851-t004:** Size of the baffles suppressing reflux.

Baffle Size	Baffle 3	Baffle 4	Baffle 5	Baffle 6	Baffle 7
Length (X)/mm	1000	1000	1000	1000	1000
Width (Z)/mm	100	50	75	75	75
Height (Y)/mm	90	90	90	70	90

Baffle 3, Baffle 4 and Baffle 5 position: 4000 mm from the taphole, 370 mm from the upper surface of the main trough. Baffle 6 position: Moved up 45 mm compared to Baffle 5. Baffle 7 position: Moved down 45 mm compared to Baffle 5.

## Data Availability

Not applicable.

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
