# Peer review of "Numerical Analysis on Erosion and Optimization of a Blast Furnace Main Trough"

_materials, 2021, doi:10.3390/ma14174851_

Round 1

Reviewer 1 Report

The manuscript materials-1347673 was reviewed. The authors numerically investigated the erosion of main trough in blast furnaces. The following comments arose during the review process:

  1. Did author consider the type of refractory material for their simulation studies? In Table 3.1.1, some specific properties of the refractory materials is presented. But  some of its initial properties such as material type, thickness, porosity and chemical composition  influence the erosion of this section in the blast furnaces. The authors are suggested to make some discussion in this regard. 
  2. The wall shear stress  is getting doubled at the end of tapping compared to its initial stage. A question here is why line 4 is more affected in comparison to other lines? 
  3. As it is discussed in the paper, the results presented are based on simulation. How do authors validate their numerical results for experimental conditions? It is an important question that determines the quality of this research study. The authors are suggested provide a clear discussion in this regard.

Author Response

Dear editor,

   Thank you for informing us the positive feedback; as well as the reviewers for quickly processing reviews of our manuscript. We feel very thanks for reviewer professional review work on our article.

Yours sincerely,

Hao Yao

State Key Laboratory of Advanced Special Steel, School of Materials Science and Engineering, Shanghai University

No.99 Shangda Road, Baoshan District, Shanghai, China

E-mail: yaohao159123@shu.edu.cn

Reviewer 2 Report

The Authors used a numerical analysis of the discharge trough operation of a real facility operating in China. To this end, They used the OpenFOAM software. They presented the model in three parts - equations that describe hot metal, refractory materials and the heat transfer of these materials. They presented the equations of conservation of mass, momentum, energy and stress on the walls of the refractory lining by hot metal. Then, they described the equations of behavior of the refractory lining of the main trough through the Fourier equations, momentum balance, strain displacement and elastic constitutive equation, and heat transfer at the border between liquid metal and refractory material. They set up calculation grids with an average size of 20 mm hexagonal cells for the main trough. They made certain assumptions (the flow of metal and slag is turbulent, the Newtonian incompressible fluid, the refractory lining is intact) and the boundary conditions described by the equations. The numerical simulations carried out using the OpenFOAM program are consistent with the simulations carried out by other researchers using Fulent. In addition, They proposed the use of baffles in the drainage channel, which can dampen turbulence and inhibit the retreat of hot metal.

Detailed comments:

line 30 and the whole text - ing[1]. There are about 4~7.5 t of hot metal and molten slag with 1733.15~1803.15 K[2] it should be like this ng [1]. There are about 4~7.5 t of hot metal and molten slag with 1733.15~1803.15 K [2]

line 191 - I propose to add „ … in Table 3.3.1 and fig. 3.3.1.”

Adopt one convention for describing equations.

Carry out a thorough editorial proofreading of the text (for example Figure 3. 1.1, Figure 4. 1.1.).

Author Response

(The authors gave the same response as above.)

Reviewer 3 Report

The manuscript entitled "Numerical Analysis on Erosion and Optimization of Blast Furnace Main Trough” numerically studied the tapping process of the main trough of a blast furnace in the east of China with the help of OpenFOAM.

The manuscript is good in quality and presents valuable information and parametric study about the tapping process of the main trough of a blast furnace. The following comments should be addressed before any further process.

Comments:

  • The English of the manuscript is good. However, it could benefit greatly from professional editing to improve technical writing and English. For examples:

             Line 15: “The results shown that”.

             Line 225: This sentence should be improved technically.

             Lines 444-446: This sentence is too long and is not clear.

  • More information about the numerical methodology used is encouraged to be provided in the abstract.
  • Line 154: Are these expressions for figures suitable? I believe it should be Figure 1. The same note for Table 3.1.1. This should be corrected through the manuscript.
  • Line 226: Are there any reasons to select these positions? Are they represent critical or maximum velocity positions?
  • Line 233: Is it possible to show these points in the Figure
  • Line 237-238: It will be clearer to show the increase in direction angle on the Figure.
  • An expression like “significantly, greatly, etc.” does not provide much information for the readers and seems to be lacking a scientific judgment and technical explanation of the phenomena. Therefore the authors should provide more specific discussions to help the reader to understand how much this improvement or this effect. This should be corrected through the manuscript.
  • Figure 4.2.3: This figure is not clear. I suggest comparing each time alone in a separate figure to be more readable.
  • Figure 4.3.1 is not clear. I suggest merging each contour distribution of the shear stress with the graph that shows the values along the runner direction.
  • Line 297: What do you mean by "very close"? How much? The authors should present this comparison with Chang et al. results.
  • Line 298: Is it the axial stress? Or what?
  • Again, the majority of the discussions and results are presented qualitatively rather than quantitatively. This has made many of the conclusions vague. An expression like “larger than, etc.” does not provide much information for the readers.
  • Line 393: Reduced by how much?
  • Line 425: More information about how did the authors investigate the fatigue life should be provided. The S-N curve should be provided and showing how to calculate the fatigue life.
  • Line 434: What do you mean by this sentence "Inhibition backflow is our second focus"?
  • Line 447: unclear expression "more obvious". The authors should avoid these kinds of expressions in the judgment.
  • Line 540: This should be section 5.

Author Response

Dear editor,

   Thank you for informing us the positive feedbacks; as well as the reviewers for quickly processing reviews of our manuscript. We feel very thanks for reviewer professional review work on our article.

Yours sincerely,

Hao Yao

State Key Laboratory of Advanced Special Steel, School of Materials Science and Engineering, Shanghai University

No.99 Shangda Road, Baoshan District, Shanghai, China

E-mail: yaohao159123@shu.edu.cn

Round 2

Reviewer 1 Report

The comments are addressed. The final decision for publication depends on the Editor.

Reviewer 3 Report

The authors addressed most of the reviewer's comments and the manuscript can be accepted for publication.